# Evaluation of Pain and Anxiety Levels After Periodontal Treatment

**DOI:** 10.3390/medicina61030464

**Published:** 2025-03-07

**Authors:** Ebru Sarıbaş, Mehmet Cudi Tuncer

**Affiliations:** 1Department of Periodontology, Faculty of Dentistry, Dicle University, Diyarbakir 21280, Turkey; 2Department of Anatomy, Faculty of Medicine, Dicle University, Diyarbakır 21280, Turkey; drcudi@hotmail.com

**Keywords:** pain, periodontal treatment, dental anxiety, stress

## Abstract

*Background and Objectives:* Nowadays, dental anxiety is one of the most common problems among the masses globally, causing individuals to avoid seeking dental treatment, which in turn leads to deterioration of quality of life related to oral health. Despite the technological advances in dentistry such that less pain is felt and high comfort performance is maintained, dental anxiety is still seen in individuals. Non-surgical periodontal therapy can be the cause of tension, anxiety, and stress. The aim of this study is to evaluate the possible relationships between pain perception and dental anxiety in individuals who underwent supragingival scaling. *Materials and Methods:* In our study, 226 individuals (114 female and 112 male) who were referred to the Dicle University Faculty of Dentistry, Department of Periodontology and who underwent supragingival scaling treatment were included. Participants were asked to complete the Modified Dental Anxiety Scale (M-DAS) and the Visual Analog Scale (VAS) to determine anxiety and pain levels and questionnaires containing information on gender, age, education level, previous dental visits, and complications. *Results:* The M-DAS score for females was significantly higher compared to males (*p* < 0.05). However, there was no statistically significant difference between genders regarding VAS scores. No statistically significant difference existed between M-DAS and VAS scores and education levels. There was a statistically significant relationship between M-DAS and VAS scores in females (*p* < 0.05). *Conclusions:* M-DAS and VAS scores in male patients did not show any statistically significant difference. Female patients exhibited higher levels of dental anxiety, and VAS scores were increased in females; M-DAS scores were also increased.

## 1. Introduction

Dental anxiety is one of the most prevalent conditions worldwide, significantly affecting oral-health-related quality of life in both children and adults. It has been reported to negatively impact individuals’ quality of life and impose a substantial burden on society. This fear causes individuals to avoid dental treatment, which has been reported to lead to deterioration of oral-health-related quality of life [1,2].

Despite decades of research, some epidemiological questions regarding dental anxiety remain unanswered. For instance, little is known about its prevalence at different stages of life, its progression over time, or the etiological pathways leading to the development of dental anxiety. Two primary reasons for this have been suggested. First, it is widely accepted that dental anxiety typically begins in childhood, thus neglecting the possibility of its onset in adulthood. Second, while most dental anxiety cases are attributed to traumatic dental experiences, there remains uncertainty about the exact causal role due to issues of recall bias and retrospective reinterpretation [3].

The impact of dental anxiety on individuals’ lives is extensive and dynamic. Its physiological effects include symptoms, such as fear response and fatigue after dental appointments, while its cognitive effects encompass a wide range of negative thoughts, beliefs, and fears. Behavioral effects go beyond mere avoidance and include changes in eating habits, oral hygiene, self-medication, crying, and aggressive behaviors [4].

Various factors contribute to the etiology of dental anxiety, including exposure to painful dental procedures, misinformation from the environment or family, traumatic dental experiences, failed dental treatments, a sense of losing control during procedures, and psychological conditions. Numerous studies have identified these factors as significant contributors to dental anxiety [5,6,7,8]. There are several anxiety scales available to evaluate dental anxiety in adults. The Modified Dental Anxiety Scale has been reported as a reliable and valid tool for use in the Turkish population [9,10,11,12]. In their study on Turkish patients, Tunç et al. found that the M-DAS possessed sufficient sensitivity. Additionally, they noted that its brevity made it a cost-effective tool for population-based studies [12].

Studies have identified pain experienced during the treatment of dental and periodontal diseases as the most critical factor in the development of dental anxiety [13,14]. Accurately measuring pain is essential, and the VAS is a common tool used to measure pain. It has been demonstrated that the VAS, a common measuring instrument in pain research and clinical practice, has linear scale characteristics for mild to moderate pain. In this scale, patients mark their perceived pain on a 10 cm horizontal line, with the data used for further evaluation [14,15]. Although the literature emphasizes the significance of dental anxiety, few studies explore its relationship with pain perception, and no consensus has been reached on this matter [13,16,17].

Given the widespread prevalence of dental anxiety and its potential impact on pain perception, there is a critical need for further exploration of the complex relationship between these factors, particularly in the context of periodontal treatment. Despite advances in pain management and patient-centered dental care, many individuals continue to experience heightened anxiety, which can negatively influence their overall treatment experience. This study aims to investigate the interplay between pain perception and dental anxiety in patients undergoing supragingival scaling, with a particular focus on gender differences. By providing a deeper understanding of these associations, our findings may contribute to the development of targeted strategies for reducing dental anxiety and improving patient outcomes in periodontal care.

## 2. Material and Methods

### 2.1. Sample Size Calculation

For this study, with a margin of error of 5%, a 95% confidence level, and an effect size of 0.66, 99.8% power was realized with 226 samples. This shows that the sample size is quite sufficient to detect this difference.

### 2.2. Study Design and Participants

This cross-sectional study included 114 female and 112 male patients who visited the Department of Periodontology at the Faculty of Dentistry, Dicle University and had completed supragingival scaling. The present study was conducted ethically in accordance with the Declaration of Helsinki (World Medical Association) and approved by the Ethics Committee of the Faculty of Dentistry, Dicle University (no: 2024/14). The research was conducted in compliance with the Declaration of Helsinki. Before the study, information about the purpose and method of the study was provided. All of the involved patients reviewed and signed the consent forms.

#### 2.2.1. Inclusion Criteria

Individuals over 18 years old who presented to the Periodontology Clinic.Individuals without dentin hypersensitivity.Individuals who had completed supragingival scaling.

#### 2.2.2. Exclusion Criteria

Individuals who refused to participate in the study.Individuals with acute periodontal abscess, pulpitis, or any acute infections.Individuals who required anesthesia due to pain.Individuals with psychiatric disorders or those taking psychiatric or pain-relieving medications.

### 2.3. Clinical Procedure

All procedures were performed by an experienced periodontist with patients sitting in the same dental chair. Supragingival scaling was performed on all patients using hand instruments without local anesthesia. After completion of supragingival scaling, all participants were asked to fill out the evaluation form.

The form included demographic variables, the Modified Dental Anxiety Scale (Figure 1), and the Visual Analog Scale for pain assessment (Figure 2).

### 2.4. Demographic Variables

The demographic data form recorded essential patient information, including gender, age, education level, and previous dental visits. Additionally, patients were asked about any history of dental complications, such as prior adverse reactions to dental treatments, prolonged post-procedural discomfort, or dental anxiety. These data were collected to assess potential correlations between demographic factors and variations in pain and anxiety levels following periodontal treatment. The gathered demographic information provided a comprehensive understanding of the patient profile, ensuring a thorough evaluation of how individual characteristics might influence treatment outcomes.

### 2.5. Modified Dental Anxiety Scale

Dental anxiety levels were assessed using the M-DAS, a validated tool designed to measure anxiety related to dental procedures. The M-DAS consists of five items, each scored on a five-point Likert scale, ranging from relaxed (score = 1) to extremely anxious (score = 5). The total score for the scale can range from 5 to 25, with higher scores indicating greater levels of dental anxiety. Participants were asked to respond to questions regarding their feelings about dental visits, including their level of anxiety before and during treatment, as well as their concerns about specific dental procedures. The total scores obtained from each item were summed and used for the evaluation of overall dental anxiety levels. This scale was chosen for its ease of application, reliability, and effectiveness in identifying patients with varying degrees of dental anxiety. The M-DAS scores were analyzed in relation to other study variables, such as pain perception and demographic characteristics, to explore potential associations between anxiety levels and post-periodontal treatment experiences.

### 2.6. Visual Analog Scale

Pain perception was assessed using the VAS, a widely recognized and reliable tool for measuring subjective pain intensity. Patients were asked to indicate their level of pain on a 10 cm horizontal VAS, where 0 represented ‘no pain and discomfort’ and 10 corresponded to ‘the worst possible pain and discomfort’. The VAS was administered immediately after the periodontal treatment and at predetermined time intervals to evaluate changes in pain perception over time. Patients were instructed to mark a point on the scale that best reflected their pain intensity at that moment. The distance (in centimeters) from the zero point to the patient’s mark was measured and recorded as the pain score. This method allowed for a quantitative and individualized assessment of pain levels, minimizing potential bias associated with verbal pain descriptors. The recorded VAS scores were analyzed in relation to demographic variables, anxiety levels (M-DAS scores), and procedural factors to determine potential associations between pain perception and patient characteristics following periodontal treatment.

### 2.7. Statistical Analysis

The GPower3.1 package program was used to calculate the power value. The data collected in this study were analyzed using licensed IBM SPSS version 21. The normality of the data distribution was assessed using the Shapiro–Wilk or Kolmogorov–Smirnov tests, depending on the sample size. Differences between groups were analyzed using the Mann–Whitney U and Kruskal–Wallis H tests for non-normally distributed variables. Post hoc tests were conducted to determine which groups exhibited significant differences when relevant. Correlation tests were applied to evaluate the relationships between variables according to their measurement levels, and *p* < 0.05 was considered statistically significant.

## 3. Results

### 3.1. Demographic and Clinical Characteristics of the Study Population

Our study included a total of 226 participants, consisting of 114 females and 112 males. In total, 50.44% of the participants in the study were female, and 49.56% were male. In terms of educational level, the highest proportion (34.96%) was university graduates. A notable 96.9% of the participants had previously visited a dentist, and 18,58% experienced complications following dental treatments (Table 1).

The average age of the individuals in the study was 32.95. The mean M-DAS (Modified Dental Anxiety Scale) score was 11.13, and the mean VAS (Visual Analogue Scale) pain score was 5.03 (Table 2).

### 3.2. Summary of Age, M-DAS, and VAS Measurements

As per the study’s findings, the relationship between M-DAS and VAS was statistically significant in female patients (*p* < 0.05), indicating that pain perception significantly influences anxiety levels in women. In male patients, no statistically significant relationship (*p* > 0.05) was found, meaning pain perception does not strongly affect their anxiety levels (Table 3).

### 3.3. Comparison of M-DAS and VAS Scores by Gender Using the Mann–Whitney U Test

There is no statistically significant difference between M-DAS and VAS scores and education levels (*p* > 0.05) (Table 4).

### 3.4. Comparison of M-DAS and VAS Scores Across Education Levels Using the Kruskal–Wallis H Test

The Kruskal-Wallis H test was performed to examine the differences in M-DAS and VAS scores across different education levels. The results showed no statistically significant difference in M-DAS scores among education levels (H = 3.57, *p* = 0.467). Similarly, VAS scores did not significantly differ based on education level (H = 1.178, *p* = 0.882). Although the mean ranks varied slightly, with higher M-DAS scores observed in the illiterate and primary-middle school groups, these differences did not reach statistical significance. Thus, education level does not appear to have a significant impact on dental anxiety or pain perception in this study population (Table 4).

### 3.5. Spearman Correlation Analysis of M-DAS and VAS Scores in Female, Male, and Total Groups

There was a statistically significant relationship between M-DAS and VAS scores in females (*p* < 0.05, r = 0.301). As VAS scores increased in females, M-DAS scores also increased. No statistically significant relationship was found between M-DAS and VAS scores in males (*p* > 0.05). When considering all participants, a statistically significant relationship was observed between VAS and M-DAS scores (*p* < 0.05, r = 0.152) (Table 5, Figure 3).

### 3.6. Scatter Plot of VAS and DAS Scores with Regression Lines for Female and Male Patients

The regression analysis examining the relationship between VAS scores and M-DAS scores is presented in the figure. Separate scatter plots were generated for female and male patients, with a fitted regression line for each group. For both female and male patients, the regression equation y = 10.39 + 0.15x suggests a weak positive association between pain perception (VAS) and dental anxiety (M-DAS). The slope value of 0.15 indicates that for each unit increase in VAS score, the M-DAS score increases by only 0.15 points, suggesting a minimal effect. While the trend appears similar for both genders, the correlation analysis revealed that this association was statistically significant in female patients (*p* = 0.001) but not in male patients (*p* = 0.472). This finding suggests that pain perception may influence dental anxiety more strongly in females than in males.

## 4. Discussion

The findings of this study demonstrate that periodontal treatment significantly impacts patients’ pain and anxiety levels, with variations depending on the type of procedure performed and individual patient factors. The results align with previous research suggesting that invasive periodontal interventions, such as scaling and root planing, may induce transient discomfort and heightened anxiety in certain patients. However, our data highlight the critical role of pre-treatment counseling and pain management strategies in mitigating these effects. Interestingly, while pain levels generally decreased within the first 48 h post-treatment, anxiety levels appeared to be more influenced by patients’ psychological predispositions and perceptions of the procedure. These findings underscore the complex interplay between physical and emotional responses to periodontal care and emphasize the need for tailored approaches to enhance patient comfort and overall satisfaction.

Dental anxiety is recognized as a state of worry and a feeling of loss of control, often linked to the sense that something dreadful might happen during dental treatment [2]. The Dental Anxiety Scale, developed by Corah, is widely used in epidemiology and clinical research and is claimed to have short yet robust psychometric properties. M-DAS was developed by adding a question about anxiety related to oral injections to the Corah Dental Anxiety Scale. Humphris et al. validated the reliability and validity of the M-DAS in their study, showing that it provides norms for both phobic and non-phobic subjects [19].

Pain, on the other hand, is an unpleasant and undesirable sensation for many individuals. The mere thought of experiencing dental pain can influence patients’ decisions to accept treatment. Initial periodontal treatment includes supragingival subgingival debridement and is the first step of all periodontal treatments [20]. This treatment is the most common procedure used to treat periodontal disease, and this procedure can be perceived as painful [21]. VAS is a simple, reliable, and valid pain assessment tool used to evaluate dental pain [22]. It has also been utilized in previous studies to assess pain resulting from periodontal treatments [23,24,25].

VAS and M-DAS were preferred in our study because they can be easily adapted to many study types, they are reliable and valid, and they have advantages in terms of comparison with the results in the literature. According to the findings of our study, female patients exhibited higher levels of dental anxiety, and a positive correlation was found between VAS and M-DAS scores in women. Similarly to our results, the literature has shown that women exhibit more prevalent anxiety in any dental treatment, regardless of which anxiety scale is used [26,27,28]. It is stated that this may be due to the fact that women can express their anxieties more easily than men [26].

Akçalı et al. reported that while anxiety levels in women were higher than in men, the perceived pain levels were found to be similar. They suggested that anxiety alone may not be the sole factor influencing perceived pain levels [13]. In our study, no significant differences were found in M-DAS and VAS scores in relation to education levels. In line with our results, some studies have identified education level as a factor affecting patients’ levels of dental anxiety [26,29,30,31]. However, some studies have reported that higher education levels are associated with greater dental anxiety in females [6,32]. There is no complete clarity in the studies showing the relationship between dental anxiety and education level.

Methods used to assess anxiety levels can yield different results across populations. The cultural background of a society may influence the content of the anxiety-inducing event, its interpretation, and the response given [33]. Variations among studies may be due to differences in study designs, as well as patient-related variables, such as age, race, psychological factors, and previous dental experiences.

In recent years, numerous studies have been conducted to reduce dental anxiety. Razavian et al. reported that the use of natural and herbal remedies, such as lavender mouth drops, helped patients feel less anxious [18]. Jaiswal et al. aimed to evaluate the effectiveness of virtual reality during tooth extraction. Anxiety levels were assessed using the M-DAS, and they reported that virtual reality effectively reduced anxiety without significantly affecting heart rate [34]. Iglesias-Rodeiro et al. compared intravenous and inhalation conscious sedation (midazolam and sevoflurane) for reducing dental anxiety during oral surgical procedures. They reported that both sedative agents were perceived as satisfactory and recommended their use [35]. Aardal et al. stated that patients with severe dental anxiety may require dental treatment under general anesthesia (GA). They emphasized that patients with severe odontogenic conditions should have access to treatment under GA if the procedure cannot be performed otherwise [36]. These approaches are considered promising in reducing dental anxiety.

The most important behavior associated with reducing anxiety is the dentist’s explicit commitment to preventing pain. Other dentist behaviors, such as acting friendly, remaining calm, showing empathy, and providing emotional support, have been reported to be closely related to patient satisfaction [37]. Fardal et al. reported that highly anxious patients’ anxiety levels decreased during periodontal treatment [38]. In our study, the rate of patients who had previously visited a dentist was found to be as high as 96.9%. We believe that different results could be obtained in patients visiting a dentist for the first time and that future studies with larger samples would contribute to the literature.

When forming the patient groups in this study, we ensured that there were no significant differences between genders. The simultaneous assessment of dental anxiety and pain levels, along with the meticulous application of inclusion criteria, add value to this study. Based on our findings, we believe that dentists should closely observe their patients and be aware of their anxiety, taking precautions against all situations that may cause anxiety or pain to effectively manage anxiety. We consider our results to be valuable and believe they will contribute to scientific developments.

Despite the strengths of our study, it has some limitations. Although the study design was as meticulous as possible, there are some limitations to this study. Previous studies have found that individuals with low socioeconomic status have significantly higher rates of dental fear compared to individuals with high socioeconomic status. The socioeconomic status of the patients was not recorded in our study. Another limitation of this study is that participants were recruited from only one center and only M-DAS was used in the evaluation of dental anxiety.

Future research on dental anxiety should focus on a multidimensional approach that integrates psychological, behavioral, and physiological factors influencing patient responses to periodontal treatment. Longitudinal studies examining the long-term effects of periodontal procedures on anxiety levels could provide valuable insights into adaptive mechanisms and coping strategies. Additionally, incorporating neuroimaging techniques and biomarker analysis may help identify objective indicators of dental anxiety, enabling the development of targeted interventions. Exploring the role of demographic factors, such as age, gender, education level, and previous dental experiences, will further enhance personalized treatment approaches. Furthermore, investigating the effectiveness of anxiety-reducing techniques, including cognitive–behavioral therapy, virtual reality exposure, and patient education, could contribute to improved patient outcomes. By integrating these strategies, future research can help refine clinical protocols to enhance patient comfort and reduce anxiety associated with periodontal treatment.

## 5. Conclusions

This study highlights the significant impact of periodontal treatment on patients’ pain and anxiety levels, emphasizing the importance of understanding and addressing these factors to improve patient experiences. While pain was found to be transient and manageable with appropriate post-treatment care, anxiety levels were influenced by both the nature of the procedure and individual psychological factors. These findings underscore the need for tailored pain management strategies and effective communication to alleviate anxiety and enhance patient satisfaction. The findings of the present study indicate that female patients exhibited higher levels of dental anxiety, and, as their pain levels increased, their dental anxiety levels also increased. In general, regardless of gender, it was found that as dental anxiety increased, perceived pain also increased. Future studies focusing on long-term psychological outcomes and the effectiveness of anxiety-reducing interventions could further optimize periodontal care.

## Figures and Tables

**Figure 1 medicina-61-00464-f001:**
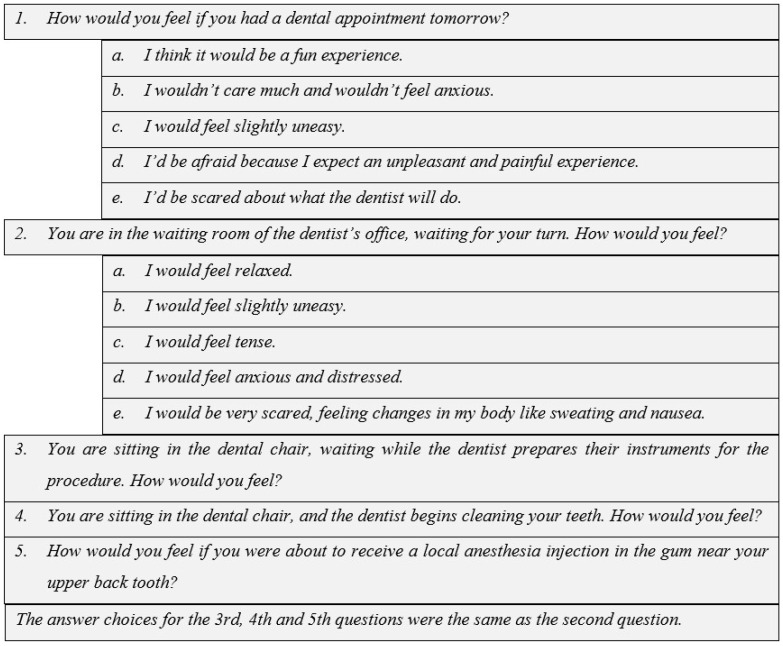
Modified Dental Anxiety Scale (M-DAS) [9,12].

**Figure 2 medicina-61-00464-f002:**
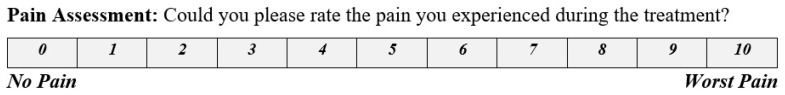
Visual Analog Scale (VAS) [18].

**Figure 3 medicina-61-00464-f003:**
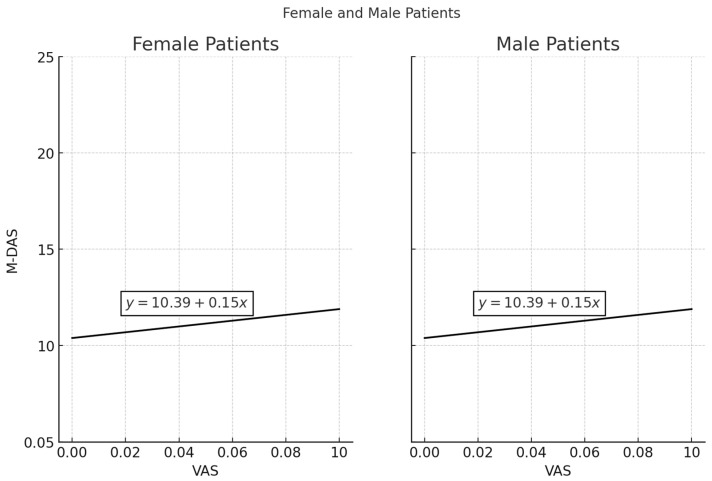
The figure presents a scatter plot illustrating the relationship between VAS (Visual Analog Scale) scores on the x-axis and M-DAS (Modified Dental Anxiety Scale) scores on the y-axis for both female and male patients. The left panel represents female patients, while the right panel represents male patients. Each point in the plot corresponds to an individual patient, displaying their respective VAS and M-DAS scores. A regression line is fitted to each dataset, with the equation y = 10.39 + 0.15x, indicating a weak positive correlation between VAS and M-DAS scores. This suggests that as pain perception (VAS) increases, dental anxiety (M-DAS) also tends to rise slightly. However, the small slope (0.15) implies that the effect is minimal. The distribution of points indicates variability in individual responses, suggesting that factors other than pain perception may play a significant role in dental anxiety levels.

**Table 1 medicina-61-00464-t001:** Demographic characteristics of study participants.

		*n*	%
Gender	Female	114	50.44
	Male	112	49.56
	Total	226	100
Education Level	Illiterate	15	6.64
	Literate	12	5.31
	Primary–Middle	45	19.91
	High School	75	33.19
	University	79	34.96
	Total	226	100
Previous Dental Visit	Yes	219	96.9
	No	7	3.1
	Total	226	100
Complication	Yes	42	18.58
	No	184	81.42
	Total	226	100

**Table 2 medicina-61-00464-t002:** Age, M-DAS, and VAS Scores.

	*n*	Mean	Median	Min	Max	SD
Age	226	32.95	31	18	68	11.93
M-DAS	226	11.13	11	5	23	4.25
VAS	226	5.03	5	0	10	2.61

**Table 3 medicina-61-00464-t003:** M-DAS and VAS scores by gender (* *p* < 0.05).

		Gender						Mann–Whitney U Test		
		*n*	Mean	Median	Min	Max	SD	Mean Rank	z	*p*
M-DAS	Female	114	12.46	12	5	23	4.24	133.79	−4.722	0.001 *
	Male	112	9.77	9	5	19	3.83	92.85		
	Total	226	11.13	11	5	23	4.25			
VAS	Female	114	4.86	5	0	10	2.17	111.14	−0.552	0.581
	Male	112	5.21	5	0	10	2.99	115.91		
	Total	226	5.03	5	0	10	2.61			

**Table 4 medicina-61-00464-t004:** Differences in M-DAS and VAS scores by educational level.

		Education Level						Kruskal–Wallis H Test		
		*n*	Mean	Median	Min	Max	SD	Mean Rank	H	*p*
M-DAS	Illiterate	15	11.93	13	5	18	4.62	127.5	3.57	0.467
	Literate	12	10.83	11.5	5	14	3.13	113.92		
	Primary–Middle	45	12.09	12	5	23	4.98	125.09		
	High School	75	11.11	11	5	22	4.31	112.97		
	University	79	10.49	10	5	19	3.78	104.68		
	Total	226	11.13	11	5	23	4.25			
VAS	Illiterate	15	5.47	5	2	10	2.13	126.97	1.178	0.882
	Literate	12	4.75	4.5	3	8	1.96	105.88		
	Primary–Middle	45	5	5	0	10	2.8	112.82		
	High School	75	5.17	5	0	10	2.69	116.17		
	University	79	4.87	5	0	10	2.63	109.96		
	Total	226	5.03	5	0	10	2.61			

**Table 5 medicina-61-00464-t005:** Correlation analysis for the relationship between VAS and M-DAS scores (* *p* < 0.05, ** *p* < 0.01).

			M-DAS
Female	VAS	r	0.301 **
		*p*	0.001 *
		*n*	114
Male	VAS	r	0.069
		*p*	0.472
		*n*	112
Total	VAS	r	0.152 **
		*p*	0.022 *
		*n*	226

## Data Availability

All details about the study can be obtained from the corresponding author.

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
