# Peer review of "Evaluation of Pain and Anxiety Levels After Periodontal Treatment"

_medicina, 2025, doi:10.3390/medicina61030464_

Round 1
Reviewer 1 Report
Comments and Suggestions for Authors
The title of the paper, “Evaluation of pain and anxiety levels after periodontal treatment,” is meaningful as a study that confirmed that gender characteristics should be reflected in order to reduce dental anxiety through analysis of the relationship between pain levels and anxiety levels and gender differences in patients undergoing scaling treatment. However, the following are the contents that need to be revised as matters to be considered in the research method.
1. In the introduction section, please clearly state the research purpose. The main result is an analysis of the relationship between current pain levels and anxiety levels, and an analysis of gender differences.
2. In the method section, ‘2.2. Study Design’ should present this research type.
The current ‘2.2. Study Design’ please content describes ‘research participants’, so it needs to be revised. In addition, please add the theoretical basis for determining the appropriate sample size.
- In addition, please describe the measurement method, judgment criteria, and reliability of the ‘Modified Dental Anxiety Scale’ and ‘Visual Analog Scale (VAS)’ by distinguishing the subtitles for the research tools.
- Please describe the data collection period, data collection method, etc. in the ‘Data Collection’ section.
3. In the Results section, Figure 3 is a repetitive description of the contents in Table 3, so it is recommended to delete it.
4. In the Discussion section, present the limitations of the study and future research directions in detail.

Author Response
Dear Editor-in-Chief,
Below we have tried to answer Reviewer 1's questions in detail. We would like to thank you and the valuable reviewer for your contributions.
Sincerely,
1. Given the widespread prevalence of dental anxiety and its potential impact on pain perception, there is a critical need for further exploration of the complex relationship between these factors, particularly in the context of periodontal treatment. Despite advances in pain management and patient-centered dental care, many individuals continue to experience heightened anxiety, which can negatively influence their overall treatment experience. This study aims to investigate the interplay between pain perception and dental anx-iety in patients undergoing supragingival scaling, with a particular focus on gender differences. By providing a deeper understanding of these associations, our findings may contribute to the development of targeted strategies for reducing dental anxiety and improving patient outcomes in periodontal care.
2. Clinical Procedure
All procedures were performed by an experienced periodontist with patients sitting in the same dental chair. After completion of supragingival scaling, all participants were asked to fill out the evaluation form. The form included demographic variables, the Modified Dental Anxiety Scale (Figure 1) and the Visual Analog Scale for pain assessment (Figure 2)
Demographic Variables
The demographic data form recorded essential patient information, including gender, age, education level, and previous dental visits. Additionally, patients were asked about any history of dental complications, such as prior adverse reactions to dental treatments, prolonged post-procedural discomfort, or dental anxiety. This data was collected to assess potential correlations between demographic factors and variations in pain and anxiety levels following periodontal treatment. The gathered demographic information provided a com-prehensive understanding of the patient profile, ensuring a thorough evaluation of how individual charac-teristics might influence treatment outcomes.
Modified Dental Anxiety Scale
Dental anxiety levels were assessed using the M-DAS, a validated tool designed to measure anxiety related to dental procedures. The M-DAS consists of five items, each scored on a five-point Likert scale, ranging from relaxed (score = 1) to extremely anxious (score = 5). The total score for the scale can range from 5 to 25, with higher scores indicating greater levels of dental anxiety. Participants were asked to respond to questions regarding their feelings about dental visits, including their level of anxiety before and during treatment, as well as their concerns about specific dental procedures. The total scores obtained from each item were summed and used for the evaluation of overall dental anxiety levels. This scale was chosen for its ease of application, reliability, and effectiveness in identifying patients with varying degrees of dental anxiety. The M-DAS scores were analyzed in relation to other study variables, such as pain perception and demographic characteristics, to explore potential associations between anxiety levels and post-periodontal treatment experiences.
Visual Analog Scale
Pain perception was assessed using the VAS, a widely recognized and reliable tool for measuring subjective pain intensity. Patients were asked to indicate their level of pain on a 10 cm horizontal VAS, where 0 rep-resented 'no pain and discomfort', and 10 corresponded to 'the worst possible pain and discomfort'. The VAS was administered immediately after the periodontal treatment and at predetermined time intervals to eval-uate changes in pain perception over time. Patients were instructed to mark a point on the scale that best reflected their pain intensity at that moment. The distance (in centimeters) from the zero point to the pa-tient's mark was measured and recorded as the pain score. This method allowed for a quantitative and indi-vidualized assessment of pain levels, minimizing potential bias associated with verbal pain descriptors. The recorded VAS scores were analyzed in relation to demographic variables, anxiety levels (M-DAS scores), and procedural factors to determine potential associations between pain perception and patient characteris-tics following periodontal treatment.
1. Figure 3 deleted.
2. Despite the strengths of our study, it has some limitations. Although the study design was as meticulous as possible, there are some limitations to this study. Previous studies have found that individuals with low socioeconomic status have significantly higher rates of dental fear compared to individuals with high soci-oeconomic status. The socioeconomic status of the patients was not recorded in our study. Another limitation of this study is that participants were recruited from only one center and only M-DAS was used in the evaluation of dental anxiety.
Future research on dental anxiety should focus on a multidimensional approach that integrates psychological, behavioral, and physiological factors influencing patient responses to periodontal treatment. Longitudinal studies examining the long-term effects of periodontal procedures on anxiety levels could provide valuable insights into adaptive mechanisms and coping strategies. Additionally, incorporating neuroimaging techniques and biomarker analysis may help identify objective indicators of dental anxiety, enabling the development of targeted interventions. Exploring the role of demographic factors such as age, gender, education level, and previous dental experiences will further enhance personalized treatment approaches. Furthermore, investigating the effectiveness of anxiety-reducing techniques, including cognitive-behavioral therapy, virtual reality exposure, and patient education, could contribute to improved patient outcomes. By integrating these strategies, future research can help refine clinical protocols to enhance patient comfort and reduce anxiety associated with periodontal treatment.
Best regards,
M.Cudi Tuncer, Professor,Ph.D.
Corresponding author

Reviewer 2 Report
Comments and Suggestions for Authors
The aim of this study is to evaluate the possible relationships between pain perception and dental anxiety in patients who underwent supragingival scaling. The topic is interesting, and the research methodology was applied correctly and carefully. Some small changes are necessary for the article to be accepted for publication:
The introduction is well-written, and the background is clear.
-
Was a power analysis conducted to determine the minimum sample size? The research was conducted correctly, the materials and methods section is well-structured, and the statistical analysis is appropriate.
-
The discussion section is too brief and should be expanded.
-
Additionally, there are numerous statements that are not supported by references, as the study currently includes only 34 citations.
-
The clinical implications of this research are unclear: how can these findings assist clinicians in their practice?
-
I recommend that the authors add a paragraph discussing the limitations of this research. The main limitation is that the sample consists entirely of individuals from the same geographical area and the same university. This could introduce a significant bias.
I believe that with minor modifications, the article could be accepted for publication.
Author Response
Dear Editor-in-Chief,
Below we have tried to answer Reviewer 2's questions in detail. We would like to thank you and the valuable reviewer for your contributions.
Sincerely,
1. For this study, with a margin of error of 5%, 95% confidence level and an effect size of 0.66, 99.8% power was realized with 226 samples. This shows that the sample size is quite sufficient to detect this difference.
GPower3.1 package program was used to calculate the power value.
‘‘In recent years, numerous studies have been conducted to reduce dental anxiety. Razavian et al. re-ported that the use of natural and herbal remedies, such as lavender mouth drops, helped patients feel less anxious [33]. Jaiswal et al. aimed to evaluate the effectiveness of virtual reality during tooth extraction. Anxiety levels were assessed using the M-DAS, and they reported that virtual reality effectively reduced anxiety without significantly affecting heart rate [34]. Iglesias-Rodeiro et al. compared intravenous and inhalation conscious sedation (midazolam and sevoflurane) for reducing dental anxiety during oral surgical procedures. They reported that both sedative agents were perceived as satisfactory and recommended their use [35]. Aardal et al. stated that patients with severe dental anxiety may require dental treatment under general anesthesia (GA). They emphasized that severe odontogenic conditions should have access to treat-ment under GA if the procedure cannot be performed otherwise [36]. These approaches are considered promising in reducing dental anxiety.’’ The paragraph in quotation marks was added to the discussion section. All of these references were current studies from 2025.
The most important behavior Limitations
‘‘Despite the strengths of our study, it has some limitations. Although the study design was as meticulous as possible, there are some limitations to this study. Previous studies have found that individuals with low socioeconomic status have significantly higher rates of dental fear compared to individuals with high soci-oeconomic status. The socioeconomic status of the patients was not recorded in our study. Another limitation of this study is that participants were recruited from only one center and only M-DAS was used in the evaluation of dental anxiety.Future Research Strategy to Understanding Dental Anxiety’’ The paragraph in quotation marks was added to the discussion section.
‘‘Future research on dental anxiety should focus on a multidimensional approach that integrates psychological, behavioral, and physiological factors influencing patient responses to periodontal treatment. Longitudinal studies examining the long-term effects of periodontal procedures on anxiety levels could provide valuable insights into adaptive mechanisms and coping strategies. Additionally, incorporating neuroimaging techniques and biomarker analysis may help identify objective indicators of dental anxiety, enabling the development of targeted interventions. Exploring the role of demographic factors such as age, gender, education level, and previous dental experiences will further enhance personalized treatment approaches. Furthermore, investigating the effectiveness of anxiety-reducing techniques, including cognitive-behavioral therapy, virtual reality exposure, and patient education, could contribute to improved patient outcomes. By integrating these strategies, future research can help refine clinical protocols to enhance patient comfort and reduce anxiety associated with periodontal treatment.’’ The paragraph in quotation marks was added to the discussion section.
New references added to the article:
33. Razavian, H.; Mazaheri, M.; Shiri, A. Effect of Lavender Oral Drops in Reducing Dental Anxiety Among Patients Requiring Endodontic Treatment: A Randomised Clinical Trial. Eur Endod J 2025, 10, 66-72.
34. Jaiswal, S.; Neeli, A.; Chopra, S. Effect of Virtual Reality in Reducing Anxiety During Dental Extractions: A Clinical Trial. J Maxillofac Oral Surg 2025, https://doi.org/10.1007/s12663-024-02406-y.
35. Iglesias-Rodeiro, E.; Ruiz-Sáenz, P.L.; Madrigal Martínez-Pereda, C.; Barona-Dorado, C.; Fernández-Cáliz, F.; Martínez-Rodríguez, N. Safety and Satisfaction Analysis of Intravenous and Inhalational Conscious Sedation in a Geriatric Population Undergoing Oral Surgery. Healthcare (Basel) 2025; 13: 116.
36. Aardal, V.; Hol, C.; Rønneberg, A.; Neupane, S.P.; Willumsen, T. Who requires dental treatment under general anesthesia due to pain and severe dental anxiety? Findings from panoramic X-ray images and anamnesis. Acta Odontol Scand 2025; 84: 78-85.
M.Cudi Tuncer, Professor,Ph.D.
Corresponding author

Round 2
Reviewer 1 Report
Comments and Suggestions for Authors
I have confirmed that the authors have taken the review comments into account. However, some revisions are still needed as follows:
Please revise '2.2. Study design' to '2.2. Study design and Participants'. And add that the study type is a cross-sectional study.
In addition, the sentence 'For this study, with a margin of error of 5%, 95% confidence level and an effect size of 0.66, 99.8% power was realized with 226 samples. This shows that the sample size is quite sufficient to detect this difference. ' in '2.1. Ethics approval' should be moved to the '2.2. Study design and Participants' section.
Author Response
Dear Reviewer 1,
Thanks for your decision firstly. Thank you very much for your valuable contributions.
In according to Reviewer 1 suggestions;
1) 2.1. The subtitle of the study was changed to ''Sample size calculation''. (For this study, with a margin of error of 5%, 95% confidence level and an effect size of 0.66, 99.8% power was realized with 226 samples. This shows that the sample size is quite sufficient to detect this difference.) The expression specified in parentheses was placed in this section.
2) The subheading ''2.2. Study design'' was changed to ''2.2. Study design and participants''. It was also added that the study was a cross-sectional study. And, the paragraph about Ethics approval was moved to this section.
These changes made in the article are clearly shown as yellow areas.
Best regards,
M.Cudi Tuncer, Professor,Ph.D.
Corresponding author
